# Smart City Actions Integrated into Urban Planning: Management of Urban Environments by Thematic Areas

**Elizeu de Albuquerque Jacques [1,\*], Alvaro Neuenfeldt Júnior [1], Sabine de Paris [2], Matheus Binotto Francescatto [1]** and **Raquel Francieli Bilhalva Nunes [1]**

1  Production Engineering Graduate Program, Federal University of Santa Maria (UFSM), Av. Roraima, nº 1000, Santa Maria 97105-900, Brazil; alvaro.junior@ufsm.br (A.N.J.); matheus.francescatto@acad.ufsm.br (M.B.F.); raquel.nunes@acad.ufsm.br (R.F.B.N.)

2  Architecture, Urbanism and Landscaping Graduate Program, Federal University of Santa Maria (UFSM), Av. Roraima, nº 1000, Santa Maria 97105-900, Brazil; sabine.paris@acad.ufsm.br

\*  Correspondence: elizeu.jacques@acad.ufsm.br

**Abstract:** Over the years, the imbalance between population needs and urban spaces' organized development has been accentuated by increasing urbanization. The implementation of smart city actions began in the 1990s, with the development of integrated solutions in cities, harmonizing social, environmental, and economic aspects. This research measures the impact of thematic areas on smart cities' management performance. The multi-criteria Decision Making Trial and Evaluation Laboratory (DEMATEL) method was used to identify the interdependent relations between smart cities' thematic areas, structuring a diagram of cause-and-effect relations using threshold quantification values. Considering their degree of importance in smart city management, the thematic areas of technology and innovation, living environment and infrastructure, education and training, and governance and engagement are highlighted. For the degree of influence, the most influential thematic areas are coexistence and reciprocity, living environment and infrastructure, entrepreneurship, and healthcare. Also, the cause-and-effect analysis identified governance and engagement, education and training, and mobility as central thematic areas for smart city management. Finally, the research construct was developed by ranking the thematic areas' performance in urban space planning.

**Keywords:** innovation; technology; internet of things; Industry 5.0





## 1. Introduction

Urbanization and urban growth have created challenges for people's quality of life, since the increase in population density in cities imposes a high demand for natural resources and ecosystem services [1]. Human agglomeration in urban environments causes problems related to transportation, sanitation, energy, education, and housing, as well as the creation of environmental impacts, hindering urban development. Due to restrictions on urban structures and resource availability, smarter ways are needed to manage cities' challenges and provide quality services in urban spaces, considering socioeconomic aspects and sustainable urban development [2].

The global urbanization process has constantly evolved, becoming a problem due to substantial impacts of population increase in urban areas. In 1950, 30% of the world's population lived in urban areas, expanding to 55% in 2018, and expected to increase to 70–75% by 2050 [3]. Urban growth usually occurs in an unplanned way, failing to meet population needs and resulting in urban centers' disorderly development. Conventional planning tools and methods cannot cope with the urban systems' complexity, despite being necessary to understand current urban systems' problems, the inhabitants' needs, and new solutions and concepts on an economic, social, and environmental scale [4].

Smart cities, through technological solutions, are concerned for citizens' well-being, especially related to education and knowledge search infrastructure, quality, and urban

services' interactivity, mobility, and security [5]. For [6], developing smart city actions and applying the internet of things (IoT) to an urban context are potential strategies for the management and planning of information and communications technologies' demands. Technologies used to improve urban management, with an emphasis on energy efficiency, urban mobility, environmental management, and public safety, have contributed to the management of sustainable solutions in cities. For technological innovations, the IoT has emerged as a main driver of smart city initiatives. IoT-enabled systems catalyze transformations, promoting improvements for human life in urban spaces [7].

The term smart city was introduced to the scientific community by [8], highlighting technological contributions to the creation of sustainable solutions through public and private sector initiatives, emphasizing the integration of social, environmental, and economic aspects to improve quality of life and socioeconomic development in urban environments. As cities continue to face urban growth challenges, to improve quality of life, urban stakeholders are considering implementing sustainable solutions with innovative technology support associated with reduced emissions, a safer environment, and cleaner cities. Smart city actions require technological tools, as well as appropriate governance strategies, structures, and standards, derived from urban stakeholders' experiences and knowledge, as well as other cities' successfully implemented actions [9].

Recent smart city trends show the need for integrated and synergistic decision-making between sectoral city areas to choose applications and solutions aimed at the main urbanization challenges, looking to integrate city planning and management [10]. The identification of thematic areas is essential for mapping and diagnosing actions to be implemented in smart cities' planning and development.

This research measures the impact of thematic areas on smart cities' urban planning performance. In urban spaces' management, it is essential to understand the relations of influence between thematic areas to structure actions of urban planning [11]. Considering the development of smart city actions, implementing strategies by thematic area is crucial for creating new perspectives between government, society, companies, and educational institutions, all searching for an innovative environment.

Smart city actions are developed around six fundamental pillars: smart living, smart economy, smart governance, smart environment, smart energy, smart communication, and transportability [12]. This research considers the smart cities' essential pillars, verifying trends and technological alignments to systematize thematic areas for implementing a management model focused on urban spaces' social, environmental, and economic aspects.

Also, smart cities are constituted as innovative urban spaces, characterized by the construction of infrastructure solutions, enabling energy optimization, improvements in urban mobility, and minimization of environmental waste generation. The actions are based on concentrating efforts on urban planning, from collaboration among people and organizations [13,14]. This theme has socioeconomic relevance, in line with [15], given the proposition of implementing smart initiatives through innovative management of urban spaces and the development of new businesses in the local economy, especially through startups using technology as basis for their operations. Using technologies and strategies focusing on improving quality of life and resource management efficiency aims to match the increase in urbanization, providing opportunities for systemic and integrated management of human and material resources in urban environments.

The article is structured in five sections: Section 1 presents the thematic context; Section 2 covers the literature review; Section 3 discusses the methodology implemented; Section 4 shows the results, analysis, and discussion; and Section 5 describes the research conclusions and considerations.

## 2. Literature Review

Incessant population growth and urbanization have intensified the development of actions aimed at reconciling citizens' lifestyles with environmental and governance actions. The application and innovation of communications technologies played a vital role in

transforming traditional urbanization into a more intelligent and comfortable space for citizens [16]. Today's urbanization requires strategies and planning for the modernization of urban life, through advanced information and communications technology solutions aimed at digital technology infrastructure for cities [17].

Smart sustainable urbanism involves urban intelligence functions' development as an advanced form of decision support, representing new conceptions of how smart cities operate and use new forms of urban simulation, as well as optimization and forecasting methods [5]. The growing interest and the need to solve challenges related to urbanization provide opportunities for private and public investment in the development and implementation of technology, which requires the effective participation of researchers and companies, in an integrated and associated manner, searching for technological implementations by subject area [18]. The urban population performs actions with a direct impact on climate change, requiring strategies to create a new quality in urban spaces, which not only maintains environmental sustainability but also improves quality of life and environmental urbanization [19].

A smart city is associated with a better expected quality of life in urban environments through the relationships between local entities, companies, and citizens, where the city develops the capacity for urban interaction, providing intelligent services that aid citizens' coexistence and interactions with urban spaces [20,21]. The development of technological resources and applied tools encourages collaboration and communication between different smart city entities, helping to promote innovation and the construction of collaborative and creative application solutions for areas including education, health, energy, industry, environment, and security [22].

The actions integrated into urban spaces stimulate service innovation in health, transportation and parking, surveillance and maintenance of public areas, cultural patrimony preservation, and urban waste collection. Communication technology solutions integrated with the IoT in the urban context generate better-quality services for citizens, collaborating with action development for human interactivity and digital service expansion [6].

Innovative actions and strategies aimed at urban environmental sustainability are evident in European cities. For example, in Santander, Spain, more than 20,000 sensors have been installed, allowing interaction via an app (SmartSantander). The users can see the locations of buses, cabs, and police cars in real time, as well as traffic, tourist, and public space usage information. Access is public, with connection via cell phone or computer with internet access, and with augmented reality available [23]. In London, England, strategic urban mobility actions were developed, with public transport being a reference. The platform WebCAT allows for checking the means of transport available, as well as monitoring travel times, both in the central region and in peripheral parts. Other actions have been adopted through technological innovation and public–private partnerships, including the integration service between buses, trains, metro, and river lines using the Oyster Card, the bicycle rental service with Santander Cycles, and a toll system to control vehicles' circulation [24].

The search for a balance between economic development, ecological preservation, and social well-being requires advanced technologies, including artificial intelligence and the IoT, emphasizing urban environments' ecological balance and resource efficiency. Innovative actions and strategies in cities demand the development of inherently sustainable technological solutions, proposed by Industry 5.0, with the optimization of material and human resources in alignment with the United Nations' Sustainable Development Goals [25]. The technological advances of Industry 5.0 with a holistic view of spaces in society, where advanced technology is intrinsically intertwined in people's lives, combined with cooperation between different urban sectors and the use of advanced data analysis, can lead to more innovative and effective solutions in cities' resource management. The collaborative approach brings together stakeholders from different areas, including technology, environment, and social management, allowing for the development of solutions beneficial for both the people and the environment.

In urban environments, innovative strategies must be developed to achieve better performance to meet the population's needs. Therefore, it is necessary to develop structural and systemic projects, aiming to transform the urban environment into a space in which citizens, companies and public authorities can, technologically, have access to more efficient resources and services [26]. Given the research scenario, for urban spaces' management, it is necessary to adapt processes and implement city planning strategies by thematic area, in an integrated manner with technological advances, and with perspectives for actions in a network of educational institutions, researchers, companies, and public managers.

Smart cities provide the opportunity for the creation of optimized solutions to improve living conditions around the world, structured to facilitate citizens' lives, as well as the development of companies and institutions, through qualified urban services. Scientific research contemplates a systematized action located in a specific time and context, which involves a set of procedures around a theme. The methodological definition is necessary for scientific research development, as well as choosing the appropriate method to answer the research objectives, based on data collection, organization, and analysis. The methodological framework and procedures were organized into stages, as detailed in Section 3.

### 3. Materials and Methods

The methodology was structured through a systematized literature search sup-ported by a cross-sectional study, and the results were analyzed using the multi-criteria method Decision-Making Trial and Evaluation Laboratory (DEMATEL), which is a method that provides opportunities to solve the problem by generating new scientific knowledge [27]. The methodological path was based on a seven-stage methodological research diagram (Figure 1).

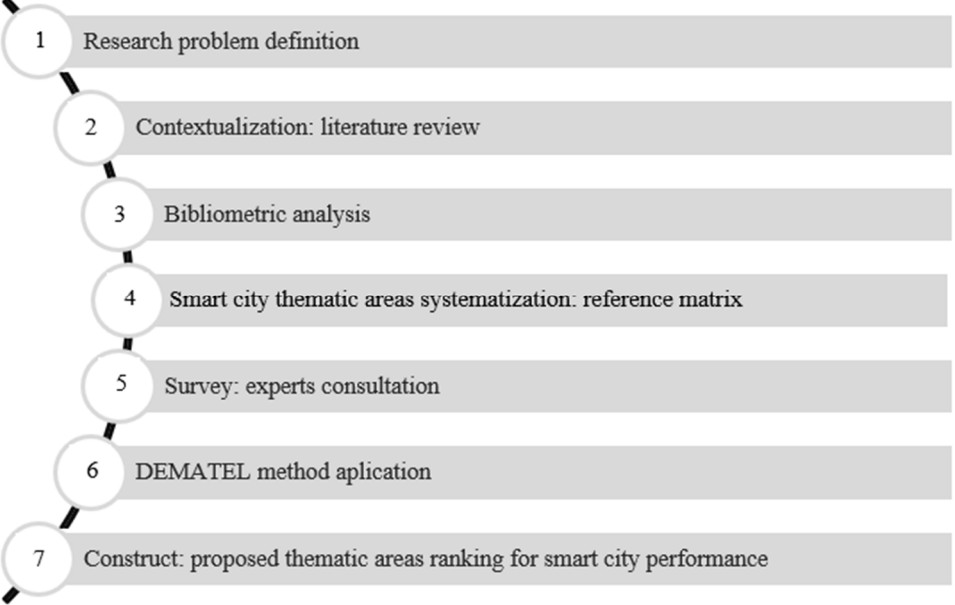

**Figure 1.** Methodological research diagram.

The first stage includes the definition of the research problem to direct and systematize the remaining stages, covering two central aspects: understanding the problem, and identifying the cause and effect of the variables addressing the problem. Also, the first stage is guided by the following research question: How do thematic areas influence the performance of smart city management in urban space planning?

For the second stage, the theoretical foundation was developed to guide the conceptual alignment and in-depth research approaching smart cities' thematic areas, by selecting articles from the Scopus and Web of Science databases (Figure 2).

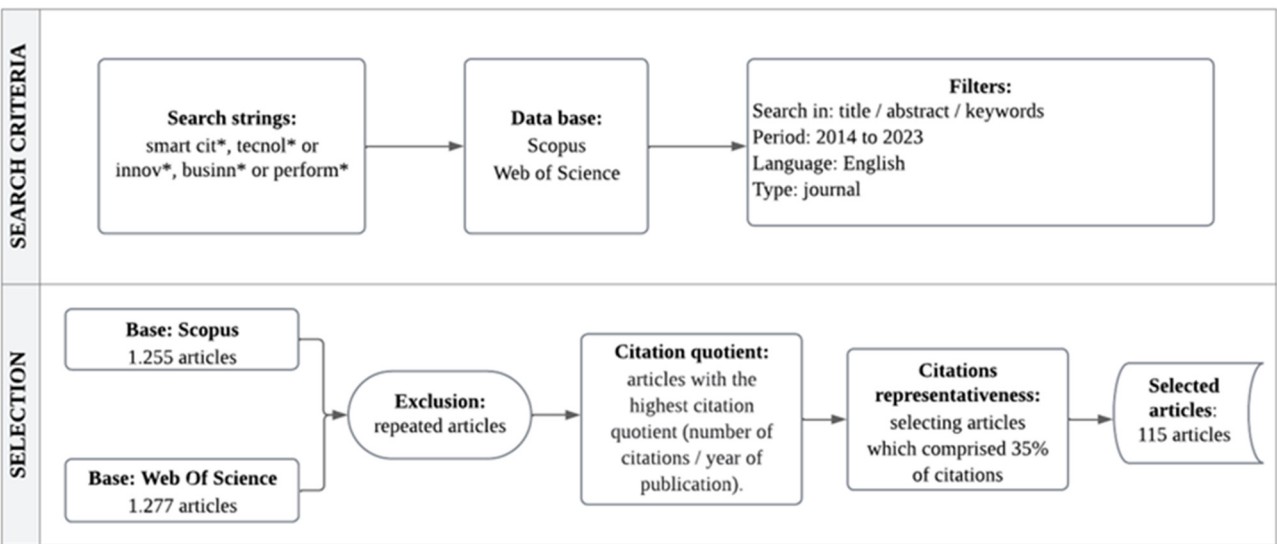

**Figure 2.** Database review protocol.

In the systematized review procedure, the search strings were defined and applied to Scopus and Web of Science, returning 2532 articles. Also, duplicate articles were excluded (956 repeated articles). For the first selection (1576 articles), the classification criterion was the database citation number, calculated by dividing the number of citations by the article publication date. Thus, the articles were ranked, and a second criterion was defined for the final selection, named citation representativeness, selecting only articles that comprised 35% of citations. Therefore, 115 articles comprised the reference base.

The information was systematized by category (author, title, source, publication year, number of citations, keywords, target audience, and thematic area) to provide publication details, structure the conceptual knowledge, and begin the bibliometric analysis. A bibliometric analysis was developed in the third stage, using the VOSviewer 1.6.20 software [28] to analyze the words' co-occurrence in the abstracts. The keywords were selected by grouping, being structured in clusters with a minimum database word frequency of 30 times. Also, terms unrelated to the area were disregarded.

Next, in the fourth stage, the most prominent words in the co-occurrence mappings were listed and, through the relationships established, the thematic areas of smart cities were identified and systematized. Also, a reference matrix with the 11 thematic areas was created, covering the segmentation of activity areas in urban space management, generating innovative solutions in city services.

For the fifth stage, an exploratory survey to collect experts' perceptions of the levels of influence between the thematic areas was created. Also, it was possible to estimate the influence levels between thematic areas using paired factors, judging the thematic areas based on their degree of influence [29]. To investigate the degree of influence between the thematic areas, a form with 11 multiple-choice questions was proposed, where each question investigated a thematic area, aiming to identify the influence received from and exerted on the other 10 thematic areas included in the reference matrix. The form was sent to Brazilian experts in four categories: smart cities researchers, municipal public managers, professionals working in the technological sector, and technology development business managers. The questions were structured with 5 possible answers: no influence (0), low influence (1), medium influence (2), high influence (3), and very high influence (4), as proposed in [11].

In the sixth stage, the interdependent relationships between the smart cities' thematic areas were identified using DEMATEL [30], allowing structures of cause-and-effect relationships to be visualized for decision-making [31]. DEMATEL evaluates a hierarchical structure based on expert opinion to determine the influence levels between thematic areas

with a relation matrix and vector calculation [32]. The hierarchy influence degree, along with the direction and intensity of the relations, is structured using a cause-and-effect relationship diagram [33].

DEMATEL is composed of five phases: In phase 1, the direct influence matrix was developed with the experts' responses on the relationships between the 11 thematic areas. An initial average matrix *A* was generated, where each element (*aij*) indicates the average degree of influence that a thematic area *i* exerts on a thematic area *j* from all *n* thematic areas. Phase 2 involved calculating the normalized influence matrix *X*, by normalizing the initial average matrix *A* by the normalization factor *k* (Equations (1) and (2)), where *k* is the normalization factor.

$$X = k.A \tag{1}$$

$$k = \frac{1}{max_{1 \le i \le n} \sum_{j=1}^{n} aij} \ , \ i, \ j = 1, 2, \dots, n \tag{2}$$

In phase 3, to construct the total relationship matrix *T*, the identity matrix *I* was initially constructed to mirror the elements of the normalized matrix, from which the total relationship matrix *T* was structured (Equation (3)), where *T* is the total matrix and *I* is the identity matrix.

$$T = X(I - X)^{-1} \tag{3}$$

When executing phase 4, rows and columns were added to the total matrix *T* to perform the calculation of vectors *D*, which included the sum per row and the vector *R*, totaling the sum per column, where *t* corresponds to element *t* of row *i* and column *j* (Equations (4)–(6)).

$$T = [\,tij\,]_{nxn'} i, j = 1, 2, \dots, n \tag{4}$$

$$D = \left[ \sum_{j=1}^{n} tij \right]_{nx} = [\,t_i\,]_{nx1} \tag{5}$$

$$R = \left[ \sum_{j=1}^{n} tij \right]_{1x} = [\,t_i\,]_{1xn} \tag{6}$$

In phase 5, based on *D* and *R*, vectors were calculated by axes, with the horizontal axis being vector (*D* + *R*) and the vertical axis being vector (*D* − *R*). The *D* + *R* value coordinates, identified on the abscissa axis, classify the thematic areas by importance level, while the *D* − *R* value coordinates, on the ordinate axis, classify the thematic areas by their level of influence on urban spaces. After calculating the vectors, the points found were identified in the cause-and-effect diagram (*D* + *R*, *D* − *R*), by quadrant, as described in [33], and then classified into central, determining, independent, and impact factors for decision-making (Figure 3).

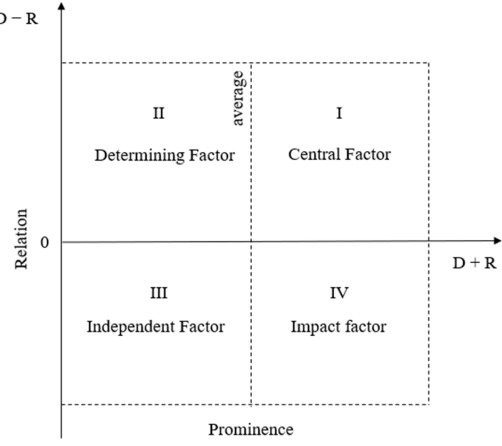

**Figure 3.** Cause-and-effect diagram structure. Source: adapted from [33].

The data were systematized and organized for analysis and interpretation, using a matrix structure and the threshold values' quantification, while considering the cause-and-effect diagram's graphical coordinates. Also, the points found were divided into four quadrants, with the $D + R$ values dividing the diagram vertically, in order of importance, and the $D - R$ values enabling the classification of points by influence level, on the horizontal axis.

For the seventh stage of the design science research, the construct was structured, including a proposal for ranking thematic areas for smart cities' performance, where the scores of the 11 thematic areas were normalized, serving as support for guiding smart cities' action planning.

## 4. Results and Discussion

The results obtained were organized to answer the research question, being systematized considering the literature review and bibliometric analysis of the smart cities' thematic areas, the data obtained from the experts, and the proposed thematic area ranking for smart cities' performance.

### 4.1. Thematic Areas for Smart City Management

Bibliometric analysis is an important technique for identifying the main thematic areas of a research field and examining how the structure of knowledge has evolved [34]. First, the database articles' keywords co-occurrence, structured in clusters (Figure 4), is presented.

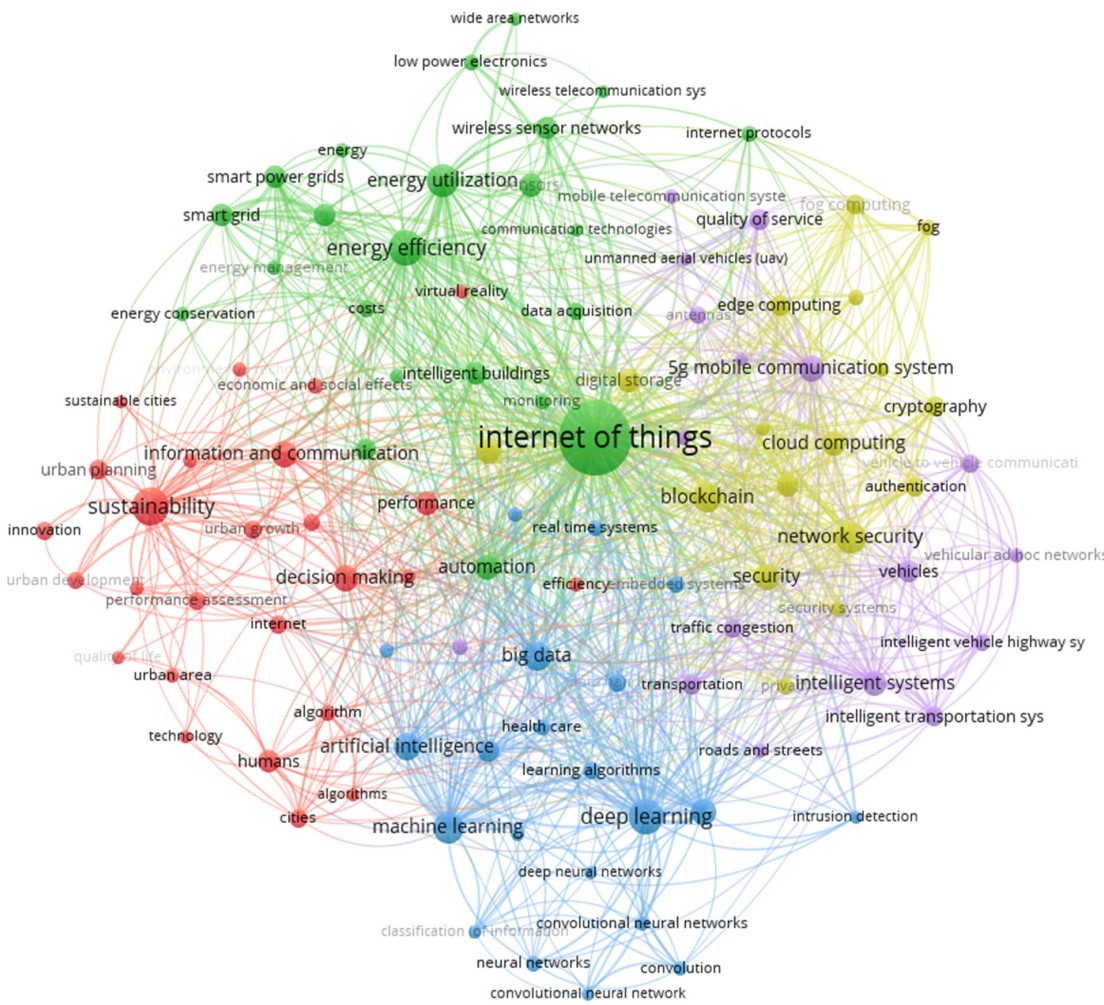

**Figure 4.** Keyword co-occurrence map.

The co-occurrence map includes 104 keywords in five clusters. Cluster 1 (red) connects words with sustainability, urban planning, and decision-making as the linking axes, with emphasis on areas of technology and innovation, and on economy and sustainable consumption. In cluster 2 (green), the interrelationships are based on IoT, energy efficiency, and energy usage, with emphasis on areas of energy, technology and innovation, and living environment and infrastructure. For cluster 3 (blue), the words learning, big data, and artificial intelligence appear, linking networked applications and services, showing the health and assistance area as a catalyst for artificial intelligence solutions. Cluster 4 (purple) approaches the interconnection between intelligent systems, communication, and transportation, with mobility as a central topic. Finally, cluster 5 (yellow) establishes connections between security, network architectures, and information management in the security and protection area.

To expand the knowledge, a word co-occurrence map from the articles' abstracts was constructed, systematized, and standardized using words with a minimum frequency of 30 times, structured into clusters (Figure 5).

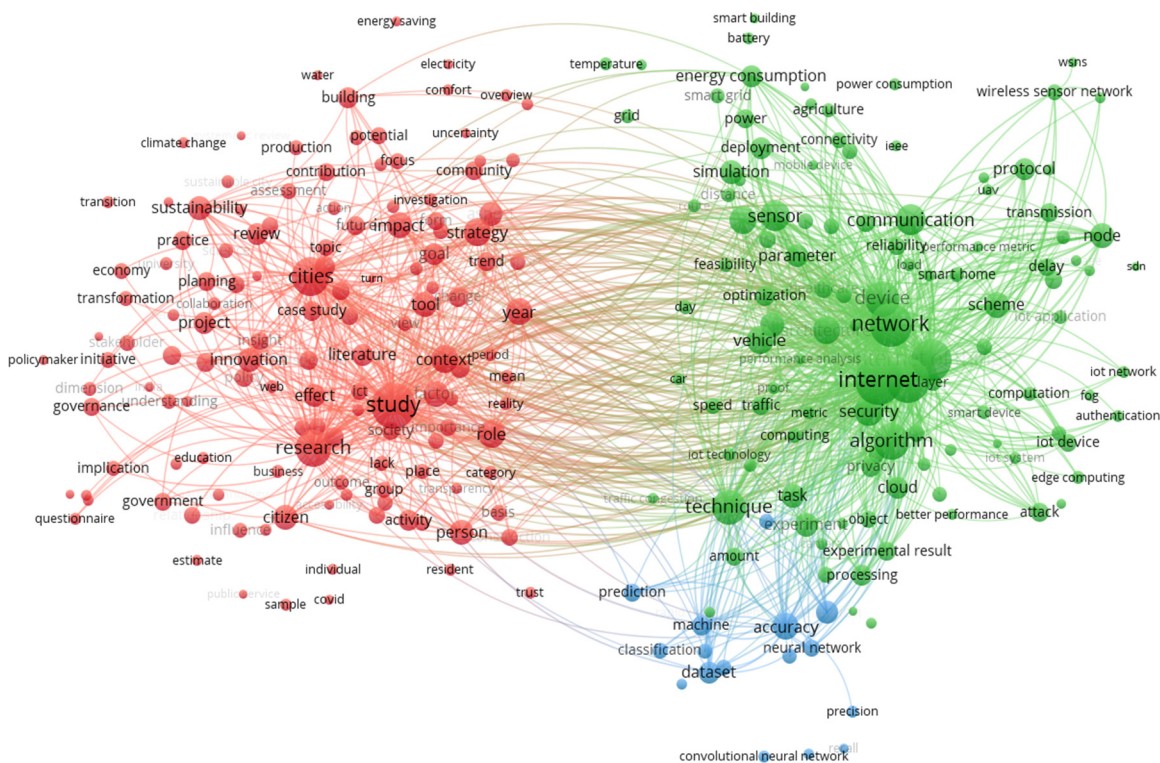

**Figure 5.** Word co-occurrence map.

The co-occurrence map includes 263 words in three clusters. Cluster 1 (red) connects the central terms study, research, cities, strategies, impacts, innovation, and governance through actions and solutions in thematic areas of governance and engagement, sustainable economy and consumption, education and training, technology and innovation, and living environment and infrastructure. In cluster 2 (green), the main relations are related to the internet, network, communication, algorithm, devices, and security, covering thematic areas of energy, safety and security, and mobility. Cluster 3 (blue) includes the terms precision, machine, dataset, and detection, highlighting actions in the thematic area of technology and innovation. The co-occurrence analysis provided an opportunity to understand the groupings and relationships between the main recurring words in applied research with an emphasis on smart cities, contributing to expanding knowledge about the perspective of smart cities' thematic areas.

With the literature review and bibliometric analysis of the words' co-occurrence, the reference matrix was structured, including the identification of 11 essential thematic areas for the development of strategic actions for the innovative management and planning of urban spaces, focused on smart cities' performance (Table 1).

**Table 1.** Smart city thematic areas reference matrix.

| Thematic Area | Description | References |
|---|---|---|
| Living environment and infrastructure | Encompasses the set of conditions and interactions essential for harmonious survival in society, in an ecologically balanced environment, focusing on quality of life and sustainability. | [18,35] |
| Coexistence and reciprocity | Values access, circulation, and support actions in public environments and spaces, with appropriate signage and assistance conditions for citizens. | [35,36] |
| Economy and sustainable consumption | Includes actions aimed at reducing social, environmental, and economic implications by optimizing and saving resources. | [18,37] |
| Education and training | Proposes actions to develop education in the region, the availability of educational institutions, and the technological structure and digital tools available to basic and higher education. | [38,39] |
| Entrepreneurship | Enterprises in operation, regional companies' numerical evolution, creative economy actions, and development of technology parks. | [40,41] |
| Energy | Proposals for optimizing energy resources, diversifying the local energy matrix, and developing strategies for alternative energy sources. | [42–44] |
| Governance and engagement | Development of management strategies using assistive technologies in public services, availability of services in electronic form, turnaround time for services, and population involvement in actions to regularize the activities developed. | [6,38] |
| Mobility | Actions and activities aimed at sustainable and efficient urban mobility to optimize travel time while reducing congestion and pollutant emissions. | [35,45] |
| Health and assistance | Innovative solutions for public health policies aimed at diagnosing and mapping population conditions, acting through procedures and automated diagnoses to optimize municipality public health conditions. | [46,47] |
| Security and protection | Monitoring actions using technological resources aimed at integrating security systems, as well as optimizing lighting control and remote management processes. | [35,36] |
| Technology and innovation | Information and communication technology applications to optimize processes, urban planning, and the development of applied solutions to improve urban spaces' quality of life. | [6,42,48] |

In smart cities, thematic areas influence each other, improving urban planning actions and quality of life. Solutions aimed at improving urban spaces are developed through the interaction and communication between thematic areas. For example, the thematic area of technology and innovation impacts several areas, since implementing the IoT in energy systems enables remote management of urban lighting, influencing the areas of energy and economy, as well as sustainable consumption. Also, real-time water quality monitoring, with information made available through mobile applications, includes interactions between the technology and innovation area and the living environment and infrastructure area.

### 4.2. Profile of Specialists Involved in the Survey

The specialists were chosen considering their academic/professional profile to obtain equal participation in four activity area categories, including the academic area, with an emphasis on research, and the professional area, working in municipal public management (mayors, secretaries, directors), technology professionals, and entrepreneurs/managers. In the data collection procedure, developed between May and August 2023, responses were obtained from 40 experts from Brazil. Table 2 shows the 40 specialists' profiles organized by their time working as a researcher/professional and their academic background.

**Table 2.** Specialists' profile.

| I - Time Working as a Researcher/Professional | | |
|---|---|---|
| Description | Quantity | % |
| Between 1 and 3 years | 3 | 7.50% |
| Between 4 and 6 years | 6 | 15.00% |
| Between 7 and 10 years | 9 | 22.50% |
| More than 10 yeas | 22 | 55.00% |
| Total | 40 | 100.00% |

| II - Academic background | | |
|---|---|---|
| Description | Quantity | % |
| Graduation | 4 | 10.00% |
| Postgraduate: MBA | 19 | 47.50% |
| Postgraduate: master's degree | 5 | 12.50% |
| Postgraduate: PhD | 12 | 30.00% |
| Total | 40 | 100.00% |

A total of 92.50% specialists had worked for more than 4 years, and 77.50% for 7 years or more, with experience in research, technology, public management, or business. The profile definition was used to aggregate applied work experience in research environments, development, and solution applications aimed at smart city actions, due to the importance of establishing influences between thematic areas in the urban space management process. Considering their academic background, 90.00% had postgraduate degrees: 47.50% in MBA courses, and 42.50% in postgraduate courses at master's and PhD levels.

### 4.3. Ranking of Thematic Areas in Smart Cities

A cause-and-effect diagram was constructed from the coordinates of the vectors $D + R$ (horizontal axis) and $D - R$ (vertical axis), integrating the relationships identified between the 11 thematic areas for smart cities. In the cause-and-effect diagram, the thematic areas were identified using the threshold value (Appendix A) and demarcated into four quadrants (Figure 6 classifies them into central, determining, independent, and impact factors within city management.

The first quadrant contains the greatest central impact factors, with the thematic areas of governance and engagement, education and training, and mobility, which are considered to be fundamental and determining factors in smart cities' performance, being identified as factors that can originate broader systemic benefits in city management. Considering the governance and engagement area, differential actions call for holistic and collaborative management, through transparency in resource application, implementation of public–private partnerships, and valuing citizens' representation and direct participation. Municipal managers are trying to implement actions and technological resources to better use tangible

resources (natural resources, distribution networks, infrastructure) and intangible resources (intellectual capital of companies and human capital) in urban space management [5].

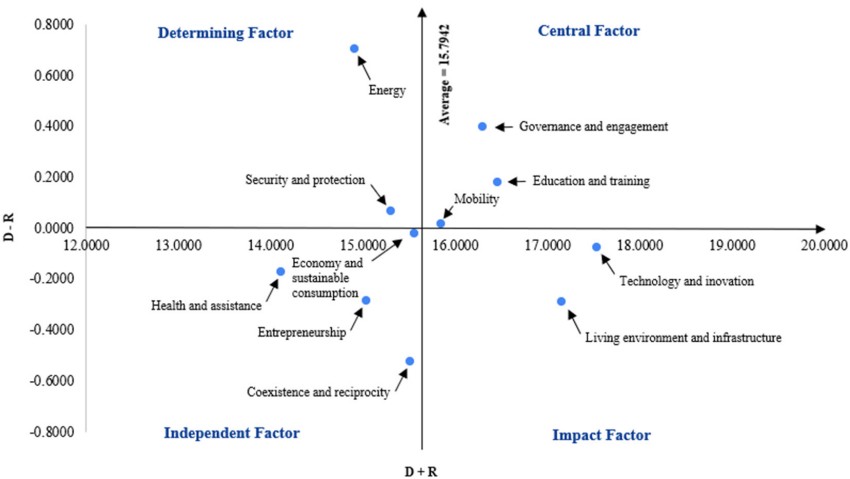

**Figure 6.** Thematic areas: cause-and-effect diagram.

The development of actions in education and training can be expanded through technological integration and the usage of digital tools in the teaching–learning process. According to [49], managing the functional needs of students is crucial to managing their training. The use of digital learning systems, assisted by teaching materials and multimedia, provides educational institutions with communication and timely feedback, as well as more precise sociability to understand learning problems through digital information. Also, digital learning systems provide students with a positive and healthy learning environment, with a close link to the institution, combined with parental monitoring of activity and learning process progress.

Mobility management is an ongoing challenge in urban areas, due to the need to implement more effective methods in city transport systems. Simulation is an important tool for diagnosing improvements in the mobility system. In the city of Herrenberg, Germany, an urban mobility simulation software test was performed to generate a better understanding of urban traffic behavior. The simulation showed how cars, trucks, buses, bicycles, and pedestrians circulate in a virtual city model, being able to test and illustrate different scenarios and visions for traffic improvements, together with reducing urban space exhaustion [4].

The second quadrant covers the thematic area of energy, as well as security and protection, which are considered to be determining factors for urban management, generating less impact but presenting a major influence on municipal management procedures and strategic decision-making aimed at promoting energy efficiency, safer urban spaces, and risk prevention. Saving energy, using alternative sources, reducing gas emissions, and monitoring energy consumption are vital in urban management. To develop smart city actions, European cities are implementing management and certification systems to control urban buildings' energy efficiency, with remote and continuous measurements in the technical systems installed, seeking to monitor energy consumption in real time and encourage the usage of alternative energy sources [50].

In urban planning actions, implementing new security and surveillance technologies is a key component in smart city management, protecting property, people, and information, preventing or detecting criminal actions, and providing residents with a sense of security [51,52]. The complexity of urban spaces requires continuous and strategic care in protecting people and infrastructure, with preventive and protective actions by public security, aiming for lower risk and safer urban environments for citizens.

The third quadrant contains independent factors: the thematic areas of economy and sustainable consumption, health and assistance, entrepreneurship, and coexistence

and reciprocity, being areas with autonomous characteristics and management, with a direct impact on the actions developed in the procedure for managing and implementing improvements in urban spaces. The economy is constantly changing due to technological innovations and digital transformation, as well as the need to comply with environmental sustainability principles. Also, the search for sustainable development compatible with economic growth and environmental preservation in urban spaces is a matter of global consensus [53]. Technological innovations and smart business models create opportunities for urban economic development and encourage environmental sustainability by saving resources and preserving the environment.

Advances in healthcare management, from the evolution of IoT and artificial intelligence tools, play a significant role in identifying health problems at an early stage, by monitoring health at a personal level 24 h a day [12]. Furthermore, in smart cities, the availability of high-speed communication networks, cloud computing, and multimedia services offers potential for action development in the telemedicine field, remote medical services, and medical data analysis and movement, contributing to quality of life. Regarding healthcare, [54] highlighted the importance of blockchain technology for the safe and rapid transfer of data to healthcare centers, which contributes to the rapid and secure transfer of patient information through smart records in hospitals.

Smart cities are perceived as ecosystems in which communities of organisms interact with their environment, helping to attract and retain specialized professionals, generating opportunities, and attracting innovative investors and entrepreneurs with financial and human capital to establish new companies in the city [55]. Entrepreneurship plays a vital role in the emergence of sustainable businesses in cities, being associated with new models for making businesses smart [56]. The smart cities aim to use urban intelligence to detect, transmit, integrate, and analyze information to enable harmonious coexistence and reciprocity in society, geared towards maintaining social ties and management supporting socially sustainable environmental principles [57].

The fourth quadrant includes impact factors with thematic areas of technology and innovation, as well as living environment and infrastructure, which receive direct influence from other areas, especially impacted by first-quadrant thematic areas: governance and engagement, education and training, and mobility. Technological innovations have revolutionized interactions in urban spaces and are providing new methods of technological integration in smart cities. Also, information and communications technology is used to improve the responsiveness, interactivity, and effectiveness of urban management services and monitoring [58]. The implementation of new technologies and appropriate management tools contributes to increasing the quality of service provision in urban spaces and socioeconomic development in cities.

In smart cities, services must improve citizens' lives. Thus, smart energy and technology play a key role in sustainable development in urban areas' living infrastructure. The living environment is the space where the compatibility and functionality of different technologies and the results created are perceived, and through which management and governance designs are impacted, adding smart actions in the human, economic, and social spheres [59].

The thematic areas were classified by level of importance from the abscissa axis position, with the areas of energy, governance and engagement, and education and training representing the thematic areas with the greatest influence on other areas, being focal areas for determining actions within cities. From the analysis of the thematic areas considering the ordinate axis, the areas with the greatest influence on urban spaces were identified, including technology and innovation, living environment and infrastructure, and education and training, which are strategic areas for targeting actions with a direct or indirect influence on other thematic areas within the scope of smart cities.

Next, the construct of the thematic areas' ranking (Table 3) was developed, based on the coordinates-normalized score identified for each thematic area, including a proposal for directing actions aimed at smart city performance.

**Table 3.** Thematic areas' ranking.

| Ranking | Thematic Area | Score | Normalized Score |
|:---:|:---:|:---:|:---:|
| 1 | Technology and innovation | 17.4575 | 10.05% |
| 2 | Living environment and infrastructure | 16.8646 | 9.71% |
| 3 | Governance and engagement | 16.7001 | 9.61% |
| 4 | Education and training | 16.6386 | 9.58% |
| 5 | Mobility | 15.8613 | 9.13% |
| 6 | Energy | 15.6097 | 8.98% |
| 7 | Economy and sustainable consumption | 15.5431 | 8.95% |
| 8 | Safety and security | 15.3753 | 8.85% |
| 9 | Coexistence and reciprocity | 14.9834 | 8.62% |
| 10 | Entrepreneurship | 14.7567 | 8.49% |
| 11 | Healthcare | 13.9466 | 8.03% |

The thematic area of technology and innovation is highlighted in first place, assembling actions resulting from improvements generated by solutions applied to urban spaces and the integration of various services provided to the population. A smart city is a concept of urban operation that uses digital information and communications technologies to make more efficient use of its infrastructure, reduce resource consumption and general costs, and meet socioeconomic objectives [60]. In second place is the thematic area of living environment and infrastructure, with the execution of actions of immediate visibility, comprising a set of essential procedures for harmonious coexistence in society, considering the spaces traveled daily by the population.

In third place is the thematic area of governance and engagement, including a set of management strategies for the various urban services essential for managing relations with society and focusing on urban spaces' collective objectives. The thematic area of education and training is in fourth place, where the relevant availability of basic and higher education and the implementation of technological pedagogical structure are highlighted, with integration of digital platforms for monitoring teaching and learning, and with a focus on connectivity and interactive activities. The importance of education goes beyond its traditional role, becoming fundamental for understanding, interpreting, and facing the challenges of modern society. Education is increasingly recognized as a critical element in enabling citizens to play more active roles in the initiatives that characterize smart cities [61].

The thematic area of mobility is in fifth place, being a major challenge for smart city management, given the high cost of implementing efficient transport models and the need to reduce carbon dioxide emissions in urban spaces. In sixth place is the thematic area of energy, which includes actions aimed at improving smart cities' energy efficiency. In smart cities, optimal energy usage, resource management, and energy optimization are the main challenges faced to improve living conditions [62].

In seventh place is the thematic area of economy and sustainable consumption, which is an area that includes financial resources integrated with the management of environmental impacts in urban actions. Smart cities' actions contribute to greater connectivity in addressing social, economic, and political issues and approaching environmental concerns, especially regarding resilience and adaptability, integrated with progress in achieving economic goals and objectives in cities [63]. In eighth place is the thematic area of safety and security, focusing on maintaining citizens' privacy and security and, with digital technology support, preventing risk situations.

The thematic area of coexistence and reciprocity is ranked ninth, approaching harmony between people and structures, circulation conditions and signage in public spaces, and

citizen assistance. The thematic area of entrepreneurship is in tenth place, promoting economic development through planning and entrepreneurial initiatives to generate new businesses and local incentives for opportunities that add value and return to cities. Finally, the thematic area of healthcare is in eleventh place, receiving influence from other smart cities areas, but having little direct influence, due to the specificities of health work. Among the applications in evidence for healthcare are technological innovations and intelligent systems, enabling the detection of incidents early, as well as optimizing care by call centers and preliminary medical recommendations, essential for rapid care in urban spaces [64].

## 5. Conclusions

Smart cities, through technologies and IoT implementation, enable urban planning development, improvements in city operations and routines, and intelligent analysis to optimize services, production, and usability in urban environments [2]. Therefore, this research contributes to the development of innovative actions and practices, in an interrelated way with technology, focusing on smart cities' thematic areas, aiming at well-being and quality of life in urban spaces. This situated action, through interactions between public management and citizens, can enhance the development of urban services solutions, optimize resources, and propose innovative sustainable actions and entrepreneurial initiatives.

The literature review and bibliometric analysis contributed to the literature mapping and the conceptual deepening in the identification and segmentation of 11 smart city thematic areas, collaborating with developing strategic actions for urban planning, since the implemented actions influence other areas. The research relevance relates to the subject's importance, analyzing a contemporary and evolving approach, contributing to innovative management with an emphasis on thematic areas, aimed at improving urban spaces' integrated management.

To identify the thematic areas' ranking, DEMATEL was used, and the data obtained from the experts' opinion were systematized for analysis and interpretation through a matrix structure and quantification of threshold values, with the degree of influence between thematic areas being used to identify cause–effect relations. The cause–effect diagram was constructed from the data collected, integrating the relations between the 11 thematic areas into four quadrants: central, determining, independent, and impact factors. In the first quadrant, as central factors, the thematic areas of governance and engagement, education and training, and mobility were identified as areas that generate systemic benefits to smart cities' management. In the second quadrant, as determining factors, with great influence on municipal management procedures and strategic decision-making, energy and safety, as well as protection, were classified. The third quadrant includes independent or autonomous factors relating to the thematic areas of economics and sustainable consumption, health and assistance, and entrepreneurship, together with coexistence and reciprocity, being areas with autonomy and growth potential in the management and implementation of innovative actions. The fourth quadrant includes thematic areas influenced by others, such as technology and innovation, as well as living environment and infrastructure, which are areas mainly impacted by first-quadrant areas, integrating potential actions to boost urban spaces' development.

After systematization through the analysis of importance and influence between thematic areas, the scores and normalization were structured to visualize the thematic areas' hierarchical ranking, with the variation's sum threshold values, which contemplate the interrelations between thematic areas, as shown in Figure 7.

Four thematic areas are highlighted. Technology and innovation are in first place, with 10.05%, impacting other thematic areas, resulting from improvements generated from applied solutions in urban service infrastructure. In second place is the living environment and infrastructure, with 9.71%, including a set of essential services for society's harmonious survival, considering the population's daily traveled spaces and actions of immediate visibility. In third and fourth place are the thematic areas of governance and engagement, with 9.61%, and education and training, with 9.58%. The governance area includes a set

of urban service management strategies for managing relations with society and focusing on urban spaces' collective objectives. For education and training, there is the relevant availability of basic and higher education, focusing on different audiences, aiming to improve living conditions and harmony with the environment. Furthermore, the remaining thematic areas also impact urban life and must be considered in the planning of urban spaces by cities in search of developing differentiated strategies.

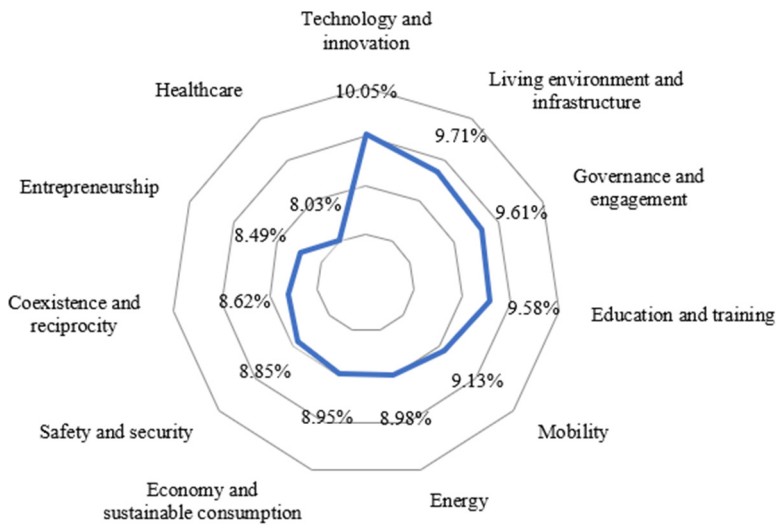

**Figure 7.** Thematic areas' hierarchical ranking diagram.

In actions aimed at economic and social development, the quest for scalability requires an understanding of strategies in the smart cities context, with organized players and organic urban environmental involvement to implement coordinated actions between public management, companies, educational institutions, and society.

Smart city management presents particularities based on urban spaces' characteristics, population growth, and constant technological evolution. This dynamism and the absence of standardized indicators applicable to the evaluation of smart cities' management performance generated limitations, hindering the comparative evaluation of actions implemented in urban environments.

Urban areas constitute one of the main sustainability issues defined by the United Nations, with the smart city concept representing a way of achieving urban sustainability objectives [65]. For further research, we suggest the evolution of this study through the definition of indicators by thematic area, while also integrating concepts of responsive cities and the contribution of citizens to the management of cities, continuing the evaluation of smart cities by thematic area. Also, studies regarding the correlation between thematic areas and sustainable development objectives from the United Nations 2030 Agenda are required, being a potential alignment for smart cities' maturity and the direction of strategic actions in the improvement of urban spaces.

**Author Contributions:** Conceptualization, E.d.A.J. and A.N.J.; methodology, E.d.A.J., A.N.J. and M.B.F.; software, E.d.A.J.; validation, A.N.J. and S.d.P.; formal analysis, A.N.J., S.d.P., M.B.F. and R.F.B.N.; investigation, E.d.A.J.; resources, E.d.A.J. and A.N.J.; data curation, E.d.A.J.; writing-original draft preparation, E.d.A.J. and A.N.J.; writing-review and editing, E.d.A.J., A.N.J., S.d.P., M.B.F. and R.F.B.N.; visualization, E.d.A.J. and A.N.J.; supervision, A.N.J.; project administration, A.N.J.; funding acquisition, E.d.A.J. and A.N.J. All authors have read and agreed to the published version of the manuscript.

**Funding:** This research was funded by the Coordination for the Improvement of Higher Education Personnel (CAPES).

**Institutional Review Board Statement:** Not applicable.

**Informed Consent Statement:** Not applicable.

**Data Availability Statement:** Data are contained within the article. If additional data are required, they can be requested by email: elizeu.jacques@acad.ufsm.br.

**Acknowledgments:** The authors would like to thank the Innovation and Competitiveness Nucleus (NIC) of the Federal University of Santa Maria (UFSM) for the incentive and opportunity to carry out this research.

**Conflicts of Interest:** The authors declare no conflicts of interest.

## Appendix A. Coordinates of Thematic Areas

**Table A1.** Threshold value coordinates of thematic areas.

| Thematic Area | $D$ | $R$ | $D - R$ | $D + R$ | Score | Ranking |
|---|---|---|---|---|---|---|
| Technology and innovation | 8.7287 | 8.8042 | −0.0754 | 17.5329 | 17.4575 | 1 |
| Living environment and infrastructure | 8.4323 | 8.7198 | −0.2875 | 17.1521 | 16.8646 | 2 |
| Governance and engagement | 8.3500 | 7.9532 | 0.3968 | 16.3033 | 16.7001 | 3 |
| Education and training | 8.3193 | 8.1371 | 0.1822 | 16.4564 | 16.6386 | 4 |
| Mobility | 7.9306 | 7.9151 | 0.0155 | 15.8458 | 15.8613 | 5 |
| Energy | 7.8048 | 7.1018 | 0.7031 | 14.9066 | 15.6097 | 6 |
| Economy and sustainable consumption | 7.7715 | 7.7923 | −0.0207 | 15.5638 | 15.5431 | 7 |
| Safety and security | 7.6876 | 7.6217 | 0.0659 | 15.3094 | 15.3753 | 8 |
| Coexistence and reciprocity | 7.4917 | 8.0151 | −0.5234 | 15.5068 | 14.9834 | 9 |
| Entrepreneurship | 7.3784 | 7.663 | −0.2847 | 15.0414 | 14.7567 | 10 |
| Health and care | 6.9733 | 7.145 | −0.1717 | 14.1183 | 13.9466 | 11 |

Legend: $D$: Summation per row of matrix $T$. $R$: Summation per column of matrix $T$. $D - R$: Coordinates of thematic areas on the horizontal axis by level of influence. $D + R$: Coordinates of thematic areas on the vertical axis by level of importance. Score: Score of the coordinates identified by thematic area. Ranking: Position occupied by thematic area based on the score.

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
