# Peer review of "Smart City Actions Integrated into Urban Planning: Management of Urban Environments by Thematic Areas"

_applsci, doi:10.3390/app14083351_

Round 1
Reviewer 1 Report
Comments and Suggestions for Authors
This manuscript discloses a comprehensive study on the influence of thematic areas on the performance of smart cities management. In my opinion, the research is well-structured, and the methodological approach is clearly outlined in the document.
Here are some suggestions to improve the manuscript:
1. I suggest improving the transition from the literature review to the methodology section, because it could include a brief summary or a concluding statement that leads into the next section.
2. The use of the DEMATEL method to identify the interdependent relationship between smart cities' thematic areas is well-explained and adds depth to the analysis. However, it would be beneficial to include more detailed explanations or examples of how these thematic areas interact and influence each other in a real-world context.
3. In my opinion, the conclusions are fine, but they should explicitly state the results obtained. I suggest the authors take the most relevant concrete results and highlight them in the conclusions of this manuscript.
Otherwise, it is a very complete, conclusive article, with an appropriate methodology, and written in a coherent manner, checked against the literature.
Author Response
Follow attached file.

Reviewer 2 Report
Comments and Suggestions for Authors
Dear Authors,
Please consider the following suggestions as for improving your manuscript.
At first, minor issues such as starting a phrase with a reference number ([6] highlight the use of communication) should be avoided
Secondly, your literature review is insufficient, and as a consequence, your Resources list as well.
Please Include to your study the Limitations and Future developments in regard to the studied area.
Best regards,
Author Response
Follow attached file.

Reviewer 3 Report
Comments and Suggestions for Authors
The paper reports on a study on the relation of thematic areas and smart city management performance.
I decided to recommend a major revision for this paper. There are some critical issues that need to be addressed.
- The manuscript claims to report on a design science research (DSR) study. However, it seems that the research design described fails to include some key elements of DSR, such as a clearly defined design artifact and its evaluation. For example, see:
Venable, J., Pries-Heje, J., & Baskerville, R. (2016). FEDS: a framework for evaluation in design science research. European journal of information systems, 25, 77-89.
- As an alternative, the manuscript could report on a systematic literature survey and a cross-sectional survey study - the results of which were analyzed with DEMATEL.
- The manuscript does not provide enough information to repeat the literature survey. For example, it does not report the used search queries, the criteria "highest citation quotient" does not have a threshold/cut-off value (also missing the justification for it), etc. The manuscript also does not report how many papers were excluded in specific steps (e.g., removing duplicates, excluded after reviewing title/abstract, etc.). Please see PRISMA guidelines for reporting structured literature surverys.
- The study also involves human participants however no ethical considerations are reported. For example, was an approval obtained from an ethics committee board prior to the study?
- The paper does not report any validity/reliability checks for the survey. For example, how did the researchers deal with the bias resulting from the potential inability of respondents to adequately infer the associations between constructs. It is highly unusual to ask the respondents to do something that we typically test with statistical methods (i.e., measure two constructs independently and test their associations with statistical methods).
Without these details it is impossible to evaluate whether the presented results have merit, and their implications.
- Eq. 1: What is the meaning of the dot (.) - product?
- Eqs. 2, 4-5: Missing subscript aij, tij.
- Eq. 5 and 6 are the same.
- Fig. 3: Check capitalization.
Good.
Author Response
Follow attached file.
